# Wi-SL: Contactless Fine-Grained Gesture Recognition Uses Channel State Information

**DOI:** 10.3390/s20144025

**Published:** 2020-07-20

**Authors:** Zhanjun Hao, Yu Duan, Xiaochao Dang, Yang Liu, Daiyang Zhang

**Affiliations:** 1College of Computer Science and Engineering, Northwest Normal University, Lanzhou 730070, China; 2018221805@nwnu.edu.cn (Y.D.); dangxc@nwnu.edu.cn (X.D.); liuy2018@nwnu.edu.cn (Y.L.); 2019221828@nwnu.edu.cn (D.Z.); 2School of Electronic and Information Engineering, Lanzhou Jiaotong University, Lanzhou 730070, China

**Keywords:** WiFi, device-free, gesture recognition, CSI, K-means, SVM, human activity recognition (HAR)

## Abstract

In recent years, with the development of wireless sensing technology and the widespread popularity of WiFi devices, human perception based on WiFi has become possible, and gesture recognition has become an active topic in the field of human-computer interaction. As a kind of gesture, sign language is widely used in life. The establishment of an effective sign language recognition system can help people with aphasia and hearing impairment to better interact with the computer and facilitate their daily life. For this reason, this paper proposes a contactless fine-grained gesture recognition method using Channel State Information (CSI), namely Wi-SL. This method uses a commercial WiFi device to establish the correlation mapping between the amplitude and phase difference information of the subcarrier level in the wireless signal and the sign language action, without requiring the user to wear any device. We combine an efficient denoising method to filter environmental interference with an effective selection of optimal subcarriers to reduce the computational cost of the system. We also use K-means combined with a Bagging algorithm to optimize the Support Vector Machine (SVM) classification (KSB) model to enhance the classification of sign language action data. We implemented the algorithms and evaluated them for three different scenarios. The experimental results show that the average accuracy of Wi-SL gesture recognition can reach 95.8%, which realizes device-free, non-invasive, high-precision sign language gesture recognition.

## 1. Introduction

As intelligent devices have been integrated into thousands of households, the traditional means of human-computer interaction can no longer meet the growing needs of users. The emergence of a series of new technologies, such as face recognition, gesture recognition, and body posture recognition, has completely updated the way of human-computer interaction. As one of the core technologies of new human-computer interaction, gesture recognition has become an active research topic [1]. Gestures are different from traditional human-computer interaction methods (such as mouse, keyboard, touch screen), because they are simple, intuitive, easy to learn, and can express ideas naturally, which can bring users a more friendly experience. Gesture recognition technology has been widely used in the smart home, sign language recognition, robot remote control, virtual reality devices [2]. There are nearly 70 million deaf and hearing-impaired people in the world. For them, sign language is a key tool for equal communication and contact with the world. Especially in specific scenarios, such as the legal needs of people with hearing impairment, cultural communication, they need to communicate with people who are not familiar with sign language. Although the intelligence of interactive technology and the emergence of sign language interpreters have solved these problems, the existing gesture recognition technology has a lack of universality, such as high price and inconvenience to carry.

The existing gesture recognition systems can be divided into three categories: computer vision, wearable sensors, and radio communication technology. Among them, the computer vision method captures the video stream of gesture movements through optical cameras, uses image processing technology to extract gesture features, and uses strict mathematical models for pattern recognition to obtain reliable recognition results. It is a relatively mature method at present, and many representative products have been born. For example, Leap’s LeapMotion [3] and Microsoft’s Kinect [4], but because of relying on optical high-definition cameras, this method does not work in low light and water fog environment, and the recognition range of gestures is limited by the shooting range of the camera, and cannot be used in some scenes where users’ privacy needs to be protected. However, gesture recognition systems based on wearable sensors generally need to be equipped with special sensors, such as accelerometer [5], gyroscope [6], air pressure sensor [7], surface electromyography sensor [8], and so on. Through these sensors, we can detect the position and speed of the hand, and realize the modeling of gestures, thus perceiving different gestures. Among them, the more famous are the virtual glove CyberGlove [9] and the wearable device Baidu Eye [10]. CyberGlove uses 22 sensors integrated on the glove to sense the joint motion of the human hand and converts the motion of the hand and finger into digital data. Baidu Eye recognizes the movements of the fingers to identify objects in the air. Although the gesture recognition technology using special sensors can directly obtain fine-grained hand and finger motion data and achieve higher recognition accuracy, due to the need for users to wear additional devices, limited to the sensing distance of the sensor and expensive deployment and maintenance costs, it cannot be used on a large scale. The gesture recognition scheme using radio communication technology mainly uses the reflection of hand gestures of radio signals for gesture recognition. According to different radio equipment and detection indicators, there are several gesture recognition methods. WiSee [11] is a gesture recognition system based on Universal Software Radio Peripheral (USRP). By analyzing the Doppler effect of WiFi signals, it realizes the recognition of nine commonly used gestures in an indoor environment. Wiz [12] uses the Frequency Modulated Continuous Wave (FMCW) technology, analyzes the time from the transmitter to the receiver according to the signal affected by gestures, and realizes user action recognition and three-dimensional gesture pointing. The radio communication technology does not require the human body to wear additional equipment, the environmental requirements are low and have high recognition accuracy, but the transmission of these radio signals requires special equipment with high cost, which is not universal and cannot be popularized in daily life.

The above systems have some problems, such as high cost, low universality, high intrusiveness, which can no longer meet the needs of gesture recognition applications to integrate into daily life, especially specific needs such as sign language gesture recognition. In recent years, the rapid development of wireless sensing technology and the ubiquitous WiFi equipment provides a new scheme to overcome these limitations. The Received Signal Strength (RSS) technology based on WiFi has been widely used in indoor positioning [13,14]. And, the RSS has made some progress in gesture recognition, the most representative of which is Wi-Gest [15] is an application system that uses hand movements to cause changes in the WiFi device RSS for gesture recognition. However, because RSS belongs to the Media Access Control (MAC) layer signal, it has inherent defects such as instability, coarse-grained and vulnerable to environmental factors, so it is impossible to identify gestures with high precision. The emergence of CSI solves this problem. As a fine-grained and stable channel feature of the Physical layer, CSI is found on commercial WiFi network cards and presents the wireless channel characteristics of the subcarrier level in the way of Orthogonal Frequency Division Multiplexing (OFDM) demodulation. Compared with RSS, it has better time stability and finer channel frequency domain information. CSI has done a lot of research in high-precision indoor positioning [16,17], human motion recognition [18,19], human behavior detection [20,21]. At present, there have been some achievements in the research of gesture recognition based on CSI. WiG [22] first uses the device-free WiFi network card and the channel characteristics of general routers to achieve fine-grained gesture recognition with high precision. Wi-Finger [23] recognizes finger gestures of digits 1–9 by establishing user gestures and CSI signal changes caused by different gestures. WiAG [24] uses changes in Channel Frequency Response (CFR) model parameters caused by human gestures to realize gesture recognition independent of position and direction. WiGeR [25] extracts CSI in commercial devices, uses the Dynamic Time Warping (DTW) algorithm to associate specific hand movements, and realizes 7 gesture movement recognitions that can control smart homes. Wi-Sign [26] uses three WiFi transceivers to associate gesture movements with CSI signal waveform changes and realizes the recognition of eight commonly used sign language gestures. WiMorse [27] uses CSI to sense subtle finger movements and achieves 95% recognition accuracy for Morse codes generated by fingers.

In this paper, we propose a gesture recognition method using amplitude and phase difference information in CSI under 5.7 GHz wireless signals, named Wi-SL, 12 gestures (all commonly used sign language actions) are recognized by low-cost and widely used commercial WiFi devices. Figure 1 shows the specific gestures selected in this paper. We make use of the different phase difference information produced by the motion of different parts of the hand, and divide the recognized gesture into two categories, namely Finger language gesture (the combined action of finger joint and hand) and Sign language gesture (the combined action of arm and hand), filter out the environmental interference in the frequency domain through the Butterworth low-pass filter, use wavelet function to smooth the CSI data to facilitate the extraction of gesture features, and use K-Means and SVM to form a low-complexity KSB classification model to achieve high-precision recognition of gestures.

The main contributions of this work are as follows:We propose a CSI-based device recognition method for sign language actions, Wi-SL. Under the wireless signal of 5.7 GHz frequency band, the amplitude and phase difference characteristics of the sub-carrier level are correlated with sign language actions to realize intelligent, high-precision, contactless sign language action recognition.We construct an efficient denoising method in the Wi-SL system and use a Butterworth low-pass filter combined with a wavelet function to effectively filter multipath components and environmental interference to ensure the accuracy of sign language recognition. In addition, we have designed a reasonable optimal carrier selection strategy that effectively reduces system computational overhead.In the data classification and fingerprint matching stage of the Wi-SL system, an efficient KSB classification model is designed. KSB uses K-means to complete the sign language action data clustering and uses the integrated learning Bagging algorithm combined with the majority voting strategy to complete the selection of the optimal SVM classifier to achieve the efficient classification of sign language action feature data.In three scenarios, the corresponding multipath effect is from strong to weak (laboratory, corridor, hall) to test the performance of Wi-SL. The experimental results show that the system is highly robust, with an average recognition rate of 95.8% in Line-Of-Sight (LOS) and 89.3% in Non-Line-Of-Sight (NLOS).

This article is organized as follows. Section 2 introduces the CSI and the background knowledge of channel feature selection and gesture recognition; Section 3 introduces the method design of Wi-SL, gives the overall process of the method, and elaborates the process in detail; Section 4 introduces the experimental scenarios settings and important experimental parameter settings, analyzes the factors affecting the experimental effect, and evaluates the comprehensive performance of the system; Finally, Section 5 summarizes the full text and provides an outlook for the future research work.

## 2. Preliminary

### 2.1. Channel State Information

In a modern wireless communication system, the carrier of the whole channel can be divided into several orthogonal subcarriers by using OFDM technology, and each subcarrier contains the corresponding physical layer information. CSI has been widely discovered as a more fine-grained physical layer feature. It not only describes the channel frequency domain properties of the sub-carrier level of the data link but also describes the combined effects caused by reflection, scattering and signal propagation energy attenuation of different indoor propagation paths [28], so CSI can be expressed as
(1)$$H(t)=\sum_{n=1}^{L}h_{n}(t)\cdot e^{-j\alpha_{n}(t)}\delta \left(\tau -\tau_{n}(t)\right)$$
where H(t) is the Channel Impulse Response (CIR) of the received signal, L is the number of all multipath components, $h_{n}(t)$ and $\alpha_{n}(t)$ are the amplitude and phase values on the *n*th path component, and $\tau_{n}(t)$ is the propagation delay of the *n*th path. After Fast Fourier Transform (FFT) processing, H(t) is represented as CFR complex matrix H at the signal receiver, which reflects the channel gain information of the subcarrier level.
(2)$$H={\left[H(f_{1}),H(f_{2}),\ldots ,H(f_{i}),\ldots ,H(f_{m})\right]}^{T},1\leq i\leq m$$

Assuming that the channel band is 5.7 GHz, the channel bandwidth is 40 MHz, and the packet time is certain, we can obtain the CSI value of 2 × 3 × 114 received by the TP-Link WDR 4310 router, then we can obtain the CSI value of 684 subcarriers in total.

### 2.2. Human Gesture Recognition Based on WiFi Signal

The movement of the human hand will change the propagation path of the wireless signal, and this change will have different effects on the received signal at the receiver. For gesture recognition, as long as the effect of human hand movement on the wireless signal is modeled, accurate gesture detection can be achieved. The effect of hand movement on wireless signal propagation is shown in Figure 2.

When the human hand moves, according to the influence on the wireless signal propagation, the propagation path can be defined as two types, one is the static path which is not affected by the hand movement, and the other is the dynamic path affected by the hand movement [29,30]. The CFR of static and dynamic paths are represented by $H_{S}(t)$ and $H_{D}(t)$, respectively, where in
(3)$$H_{D}(t)=\sum_{k\in P_{D}}\alpha_{k}(t)e^{-j2\pi d_{k}(t)/\nu }$$

In Equation (3), $P_{D}$ is all the dynamic paths affected by hand movements, $e^{-j2\pi d_{k}(t)/\nu }$ is the phase shift of the dynamic path at time *t* when the path length is $d_{k}\left(t\right)$, and ν is the wavelength of the WiFi signal. Then the total CFR at the receiver can be expressed as
(4)$$H_{Sum}(t)=e^{-j2\pi \Delta f}\left(H_{S}(t)+\sum_{k\in P_{d}}\alpha_{k}(t)e^{-j2\pi d_{k}(t)/\nu }\right)$$

In Equation (4), $e^{-j2\pi \Delta f}$ is the phase difference caused by the carrier frequency difference of receiving and transmitting equipment, in which $H_{S}(t)$ is a constant, while $H_{D}(t)$ is a dynamic variable affected by time, amplitude and phase offset, so the gesture acquisition is completed by using CSI signal, the relationship between gesture and CSI signal amplitude and phase information is established, and the recognition of human gesture can be completed by data processing, classification, and matching.

### 2.3. Channel Feature Selection and Gesture Recognition

The data in this paper are all collected on commercial routers, with two transmitting antennas Nt at the transmitter (Tx) and three receiving antennas Nr at the receiver (Rx). Due to the use of Multiple-Input Multiple-Output (MIMO) technology in WiFi, the antenna array composed of Tx and Rx will form six independent data transmission links, each of which contains 114 subcarriers, so the CSI data can be expressed as
(5)$$CSI_{Nt*Nr}=\left\{\begin{array}{c} CSI_{1,1},CSI_{1,2},\ldots ,CSI_{1,114}\\ CSI_{2,1},CSI_{2,2},\ldots ,CSI_{2,114}\\ \vdots \\ CSI_{6,1},CSI_{6,2},\ldots ,CSI_{6,114} \end{array}\right\}$$

The relationship between the CSI data link and the subcarrier amplitude is shown in Figure 3. It can be seen that the amplitude values of the subcarriers in different CSI data links have large differences (Figure 3a), but in the same CSI data link, different subcarriers have similar effects (Figure 3b). Therefore, the analysis of the signal effects of gesture actions can be accomplished by using a single subcarrier in a data link with reasonable subcarrier selection.

In recent years, most CSI human perception studies have used amplitude as the eigenvalue because it is easy to obtain. It is widely used to reflect the influence of human motion on the energy change of the channel [31], but the amplitude is very unstable and easy to fluctuate due to environmental changes, so it cannot be used as a feature of some fine-grained actions, especially in the field of gesture recognition. If only a single amplitude is used as a feature, the overall performance of the system will be degraded. Compared with the variability of amplitude, the phase difference can remain relatively stable in a certain period, and can better describe the channel frequency changes of different data streams. Therefore, the combination of amplitude and phase difference as recognition features can better reflect the impact of hand actions on wireless signals.

## 3. Wi-SL System Design

### 3.1. System Flow

We propose a contactless fine-grained gesture recognition method based on CSI, namely Wi-SL. The specific flow of this method is shown in Figure 4. Firstly, the data of sign language actions are collected in the laboratory with serious multi-path, the relatively empty corridor, and the empty hall. Then, the outliers are processed by wavelet function combined with low-pass filtering, and then the amplitude information and relatively stable phase information are extracted as mixed eigenvalues to establish the fingerprint information of sign language actions. Finally, we detect 12 different gestures in real-time in three experimental environments. At this stage, we use the KSB classification model composed of the K-means algorithm and Bagging optimized SVM classifier to classify the real-time data and get more accurate gesture recognition results.

### 3.2. Data Preprocessing

#### 3.2.1. Amplitude De-Noising

We use wireless signals with 5.7 GHz frequency to sense sign language actions. Due to the indoor multipath effect and the interference in the surrounding environment, the CSI data collected in this environment more or less contain some noises, and the CSI amplitude changes caused by the human body are in a lower frequency range [32]. The original time-domain amplitude data is shown in (a), (d) of Figure 5. These noises will reduce the detection rate of gesture recognition and the performance of the method. For this reason, we design a more reasonable denoising process, using Butterworth low-pass filter (the filter order is 5th order and the sampling frequency is 400 Hz) to filter out the environmental high-frequency interference, retaining the low-frequency data of human actions, and then combined with the wavelet function smoothing data based on db3 (the detail coefficient uses minimaxi threshold mode and scale noise, and the number of decomposition layers is 5) to facilitate the extraction of effective amplitude features.

After Butterworth low-pass filtering, the results are shown in (b), (e) in Figure 5, from which it is obvious that the high-frequency anomalous interference with fast frequency changes in the amplitude is filtered out, and the low-frequency amplitude changes caused by hand movements are preserved, and (c), (f) in Figure 5, the data are smoothed using the db3 wavelet function based on the low-pass filtering. In this way, we can effectively extract gesture feature data based on preserving data integrity, which greatly reduces the complexity of gesture data and facilitates the establishment of amplitude fingerprint information.

#### 3.2.2. Obtain A Stable Phase Difference

The phase data contained in the collected CSI, as shown in Figure 6a, are randomly distributed on [0,2π]. Because of internal errors and electromagnetic interference in the environment, they cannot be directly used as fingerprint features [33]. Therefore, we need to carry out a linear transformation of the original phase to get stable phase information.

First, the phase in the *i*-th subcarrier can be expressed as
(6)$${\hat{\omega }}_{i}=\alpha_{i}-2\pi \frac{k_{i}}{N}D_{i}+C+P_{i}$$
where ${\hat{\omega }}_{i}$ and $\alpha_{i}$ represent the original and true phases extracted from the CSI data, $D_{i}$ represents the time lag of the receiver due to sampling frequency offset (SFO), and C represents the unknown phase due to carrier frequency offset (CFO), N is the number of Fast Fourier samples, the value in the IEEE 802.11 protocol family of WiFi is 64, and $k_{i}$ is the *i-*th subcarrier index (i in the experimental environment of the 5.7 GHz band in this paper is –58 to 58), and $P_{i}$ represents some noise generated in the measurement. Because of the existence of factors such as $D_{i}$, C, $P_{i}$, we cannot extract true and stable phase information from commercial routers.

Secondly, to obtain a stable true phase, we analyze and eliminate the factors that cause errors. $D_{i}$ is the mechanical delay offset of the transceiver hardware device, which cannot be eliminated [34], and we attenuate its effect by turning the device on for a period of time to achieve stable power. Since this paper uses a commercial WiFi device, the noise $P_{i}$ generated during the measurement is very small and can be ignored by approximation. However, for the interference factor C, the linear transformation method of measurement phase proposed by Sen et al. [35] is used to eliminate the unknown phase shift caused by CFO by using the phase on the entire network spectrum, where two linearly related quantities λ and μ need to be defined
(7)$$\lambda =\frac{\alpha_{i}-\alpha_{1}}{k_{i}-k_{1}}=\frac{{\hat{\omega }}_{i}-{\hat{\omega }}_{1}}{k_{i}-k_{1}}-2\pi \frac{1}{N}S_{i}$$
(8)$$\mu =\frac{1}{n}\sum_{i=1}^{n}\alpha_{i}=\frac{1}{n}\sum_{i=1}^{n}{\hat{\omega }}_{i}-\frac{2\pi S_{i}}{nN}\sum_{i=1}^{n}k_{i}+C$$
where n is the number of subcarriers in the 5.7 GHz frequency band of this paper, and the value range is 1 to 114. According to the carrier and matrix grouping in the IEEE 802.11n protocol, the subcarrier numbers of the CSI obtained in the 40 MHz bandwidth are equally spaced, and the frequency distribution of the subcarriers is symmetrical about the central frequency, so $\sum_{i=1}^{n}k_{i}=0$, that is $\mu =\frac{1}{n}\sum_{i=1}^{n}{\hat{\omega }}_{i}+C$, makes the original phase ${\hat{\omega }}_{i}$ and $\lambda k_{i}+\mu$ difference to eliminate the random phase shift to obtain the linear stable phase ${\bar{\alpha }}_{i}$, which is expressed as Equation (7):(9)$${\bar{\alpha }}_{i}={\hat{\omega }}_{i}-\lambda k_{i}-\mu =\alpha_{i}-\frac{\alpha_{n}-\alpha_{1}}{k_{n}-k_{1}}k_{i}-\frac{1}{n}\sum_{i=1}^{n}\alpha_{i}$$

Finally, after the above linear transformation, the error factor of the phase random offset is eliminated, and the real phase information of the monitoring action is restored as much as possible. By calculating these real phases on different data links, a stable phase difference information $\Delta {\bar{\alpha }}_{i}$ can be obtained.

The original phase is processed by linear transformation, and the processed result is shown in Figure 6b. It can be seen that compared with the random distribution of the original phase dispersion, the phase after the linear transformation removes environmental noise and measures internal errors, so that the scattered data of the random distribution is concentrated in a region, forming relatively stable data. It is convenient to use phase information for feature extraction.

Through a large number of analyses of the sign language sample data in this paper, we find that under the wireless signal in the 5.7 GHz band, the effect of sign language action on the phase difference is shown in Figure 7. Among them, Figure 7a shows the influence of the Sign language gestures, while Figure 7b shows the influence of the Finger language gestures.

By comparing the phase difference information changes in Figure 7a,b, we find that when the perceptual action is the Sign language gesture, the phase difference has fewer wave transformations with smaller amplitude changes in certain intervals, while the phase difference of the Finger language gesture shows continuous and violent wave transformations. Based on this, we can use different phase difference cases to distinguish between Sign language gesture and Finger language gesture.

### 3.3. The optimal Subcarrier Selection

In this paper, in the Atheros AR9580 NIC chip and 40Mhz bandwidth channel environment, the subcarrier index sequence is from number 1 to 114, and the fine-graininess of CSI is expressed on each subcarrier in the frequency domain, with different subcarriers reflecting the different effects of sign language actions on the wireless channel. Therefore, as long as subcarriers sensitive to gestures are found, sensitive features can be selected for the feature extraction stage, thereby improving the accuracy of gesture recognition. Figure 8 shows the amplitude values of the 114 subcarriers of the sign language action in the time domain transformation relationship, blue indicates low amplitude and yellow indicates high amplitude. Based on the amplitude characteristics of these frequency domains, we used the Principal Components Analysis (PCA) to extract the optimal subcarriers for 114 subcarriers, and the processing is shown in Figure 9.

In Figure 9, subcarriers 1 to 114 carry different features of sign language actions on the communication channel. If all of these feature data are used as the training set of the classification model, it will cause great computational complexity. To this end, we set a reasonable eigenvalue matrix latent and score matrix score according to the characteristics of PCA and extract the main component of the subcarrier with the largest feature contribution rate among 114 subcarriers as the optimal subcarrier. Choosing such subcarriers can better describe the sign language actions, and greatly reduce the calculation complexity of subcarrier data. Therefore, the extraction of optimal subcarriers through such a subcarrier selection strategy can ensure the accuracy of CSI sign language fingerprint establishment to a certain extent.

### 3.4. Feature Extraction

Feature selection greatly affects the performance of the device-free gesture recognition system. In the past work, statistics such as variance, standard deviation, and median absolute deviation are usually introduced as detection metrics [36]. The selection of gesture recognition features depends on the following two factors: (1) The sensitivity of selected features to sign language action perception. (2) The stability of characteristic data against environmental change. For the sensitivity of perception, we selected the amplitude and phase difference in the CSI frequency domain, which can more sensitively reflect the impact of gestures on the wireless communication link than using amplitude or phase difference alone. To enhance stability against environmental change, we first verify the independence of the two data of amplitude and phase difference and select a more stable statistical covariance as the detection metric. The verification results of the independence of the two data are shown in Figure 10.

We selected the amplitude and phase difference data of 114 subcarriers on the same data link and verified the independence of the two data by extracting their median and covariance, and the comparison in Figure 10 shows that the mean and covariance of the phase difference are mainly concentrated around 0–5, while the mean and covariance of the amplitude are concentrated around 30–40, so that the two types of data do not interfere with each other during the subsequent classification training.

Therefore, this paper uses the combination of phase difference and amplitude to extract the normalized amplitude and phase difference covariance matrix $h^{\prime }$ and $\Delta {\bar{\alpha }}^{\prime }$ as the data features of sign language movements, and the covariance matrix can be expressed as follows:(10)$$h^{\prime }=\mathrm{cov}\left({h^{\prime }}_{1},{h^{\prime }}_{n}\right)=\left[\begin{array}{ccc} E\left[\left({h^{\prime }}_{1}-\mu_{h1})({h^{\prime }}_{n}-\mu_{h1}\right)\right] & \cdots & E\left[\left({h^{\prime }}_{1}-\mu_{h1})({h^{\prime }}_{n}-\mu_{hn}\right)\right]\\ \vdots & \ddots & \vdots \\ E\left[\left({h^{\prime }}_{n}-\mu_{hn})({h^{\prime }}_{1}-\mu_{h1}\right)\right] & \cdots & E\left[\left({h^{\prime }}_{n}-\mu_{hn})({h^{\prime }}_{n}-\mu_{hn}\right)\right] \end{array}\right]$$

$\Delta {\bar{\alpha }}^{\prime }=\mathrm{cov}\left(\Delta {\bar{\alpha }}_{1}{}^{\prime },\Delta {\bar{\alpha }}_{n}{}^{\prime }\right)==\left[\begin{array}{ccc} E\left[\left(\Delta {\bar{\alpha }}_{1}{}^{\prime }-\mu_{\Delta \alpha 1})(\Delta {\bar{\alpha }}_{n}{}^{\prime }-\mu_{\Delta \alpha 1}\right)\right] & \cdots & E\left[\left(\Delta {\bar{\alpha }}_{1}{}^{\prime }-\mu_{\Delta \alpha 1})(\Delta {\bar{\alpha }}_{n}{}^{\prime }-\mu_{\Delta \alpha n}\right)\right]\\ \vdots & \ddots & \vdots \\ E\left[\left(\Delta {\bar{\alpha }}_{n}{}^{\prime }-\mu_{\Delta \alpha n})(\Delta {\bar{\alpha }}_{1}{}^{\prime }-\mu_{\Delta \alpha 1}\right)\right] & \cdots & E\left[\left(\Delta {\bar{\alpha }}_{n}{}^{\prime }-\mu_{\Delta \alpha n})(\Delta {\bar{\alpha }}_{n}{}^{\prime }-\mu_{\Delta \alpha n}\right)\right] \end{array}\right]$ where $h^{\prime }$ is the normalized form of h, $\mathrm{cov}\left({h^{\prime }}_{i},{h^{\prime }}_{j}\right)$ is the covariance between the amplitude matrix ${h^{\prime }}_{i}$ and ${h^{\prime }}_{j}$. Similarly, $\mathrm{cov}\left(\Delta {\bar{\alpha }}_{i}{}^{\prime },\Delta {\bar{\alpha }}^{\prime }{}_{j}\right)$ is the covariance between the phase difference matrix $\Delta {\bar{\alpha }}_{i}{}^{\prime }$ and $\Delta {\bar{\alpha }}^{\prime }{}_{j}$. We use the normalized CSI amplitude and phase difference covariance to construct a hybrid fingerprint matrix U, $U=\left[h^{\prime },\Delta {\bar{\alpha }}^{\prime }\right]$.

### 3.5. KSB Classification Model

#### 3.5.1. K-Means Clustering

We select the covariance feature $\Delta {\bar{\alpha }}^{\prime }$ of the phase difference in the mixed fingerprint matrix U as the input of the sample data set, cluster the two different feature data of Finger language gesture and Sign language gesture, and find the clustering center of the two types of gesture feature data. According to the classification of actions, set k=2. First of all, two data points in the training data set are randomly selected as the initial clustering center, then the Euclidean distance between each sample in the training data set and the initial clustering center is calculated, and then the data are classified into two classes according to the different distances. Each class center is updated by calculating the average value of each class until the square error of each class center reaches the minimum and the class center does not need to be changed. Finally, the K-means is used to effectively separate the sign language and finger language features, reducing the computational complexity of the subsequent SVM classifier and improving the gesture recognition efficiency.

#### 3.5.2. SVM Classification

We use the SVM classification to complete the classification of gestures and the establishment of offline fingerprint information. According to the idea of solving SVM multi-classification in literature [37], we construct a five-layer SVM classification decision tree, as shown in Figure 11a, and we use a small amount of gesture data as the input of SVM and train with LibSVM tool [38]. A preliminary classification effect is shown in Figure 11b.

The process of using SVM classification decision tree is as follows: First, in the first layer of SVM classification tree, VERY sign language action data is taken as the target class, represented by *P*, and then other sign language action data belong to the non-target class, represented by *N*. The preprocessed CSI action data is used as the SVM training sample and $K\left(x_{i},x\right)$ is the kernel function of the training sample mapped to the high latitude space *H.* We need to find an optimal classification hyperplane to distinguish the two types of data. To find the optimal classification hyperplane, we need to solve the following constrained minimum problem:(11)$$min\frac{1}{2}{\left\|W\right\|}^{2}+\beta \sum_{i=1}^{k}\left(\xi_{i}\right)$$

The constraints are
(12)$$\{\begin{array}{l} l_{i}\left(W^{T}s_{i}+b\right)\geq 1-\xi_{i}\\ \beta >0,\xi_{i}\geq 0,l_{i}\in \left\{-1,+1\right\} \end{array}$$
where W is the direction vector separating the hyperplane,β is the penalty parameter constant, $\xi_{i}$ is the relaxation variable that allows the sample to be misclassified, $s_{i}$ is the input feature vector in the training sample set, $l_{i}$ is the training specified label, and +1 is the target class (positive sample), −1 represents the non-target class (negative sample), b is the position constant of the hyperplane, and k is the number of samples, the regression function of H can be obtained by solving this problem.
(13)$$f(s)=\mathrm{sign}\left(\sum_{i=1}^{k}l_{i}j_{i}K\left(s_{i},s\right)+b\right)$$
(14)$$K\left(x_{i},x_{j}\right)=\exp\left(-\Phi {\left\|x_{i}-x_{j}\right\|}^{2}\right)$$

In Equation (13), $j_{i}$ is the Lagrange multiplier. In Equation (14), $K\left(x_{i},x\right)$ is the Radial Basis Function (RBF), where Φ(Φ>0) is a nuclear parameter. If f(s)>0, then $s_{i}$ corresponds to VERY sign language actions. In the same way, SVM classifies other gestures and obtains the classification results, to establish offline fingerprint information of gestures. We classify the feature data of VERY and BYE by SVM. As shown in Figure 11b, there is a clear empirical calibration line that can effectively distinguish the two types of feature data. Although only a few gesture feature data are used as samples, it is sufficient to show the feasibility of its data classification.

#### 3.5.3. Bagging Algorithm Optimizes SVM Classifier

In the results of SVM classification, we find that some features overlap on the hyperplane, and a single SVM classifier cannot completely classify all gesture feature samples. For example, in Figure 11b, there are some positive and negative sample features overlap near the empirical calibration line, resulting in part of the feature samples cannot be correctly distinguished. To overcome this problem, we introduce the Bagging ensemble learning algorithm [39] and take the SVM classification model of sign language action as the basic learner of the Bagging algorithm. Making use of the characteristic that its base learner can be generated in parallel and independent of each other, and combining with the absolute majority voting strategy, we select an SVM strong classifier with the best classification effect, to achieve more effective classification of sign language feature data, and improve the performance of sign language recognition. In this paper, the pseudo-code of bagging algorithm in KSB classification model is as follows:
**Pseudo-code 1:** Bagging optimizes SVM classifier**Input:** Gesture feature training set Ui={(x1,y1),(x2,y2),…,(xm,ym)}(i=1,…,Q)
Set Base learner hq, Base learning algorithm ℑ SVC (Support Vector Machine Classifier);Set Training rounds Q.**Process:**1: for  q=1,2,…,Q  do2: hq=ℑ(U,Ubs) (Ubs represents the sample distribution generated by self-service sampling)3: end for4: //The result of sample classification of basic learner c(c={c1,c2,…,cN});5: //hqN(x) the predictive classification result of the base learner hq on the sample feature x;**6: if**∑q=1Qhqj(x)>0.5∑t=1N∑q=1Qhqt(x) then**7: Return**cj;8: otherwise, Return reject;**Output:**hcj (Strong classifier with cj result of sample classification)

Firstly, we use bootstrap sampling [40] to randomly sample the sign language action feature data constructed in this paper, and Q sampling sets are obtained by Q sampling. Then, Q independent sign language SVM-based learners are obtained by SVM Classification of these sampling sets. Finally, we vote on the classification of the sample features according to the basic learner, and if more than half of the basic learners predict the same classification $c_{j}$, then the classification is the best classification after integrated learning, otherwise, the prediction is rejected. Through this way of ensemble learning, we can find the most accurate SVM classification results of sign language, to avoid the problem that sign language samples cannot be completely distinguished, and achieve a better sign language recognition effect.

## 4. Wi-SL System Design

### 4.1. Experimental Configuration

To verify the feasibility of the Wi-SL method used in actual scenarios, the transmitting and receiving devices in this article use a TP-Link WDR4310 router that supports the IEEE 802.11n protocol and has an Atheros AR9580 network card chip. The specific hardware equipment is shown in Figure 12, use Openwrt technology to update the router firmware, and install Atheros CSI-Tool [41], so that the CSI data can be obtained on the commercial router. The transmitter sends CSI data to the receiver through the Wireless Local Area Networks (WLAN) port and the physical address of the receiver, and the receiver receives the CSI data by connecting the wireless hotspot of the transmitter. The channel bandwidth is adjusted by setting the routing kernel parameters, and the transmitter and receiver bandwidth is set to 40MHz to obtain the wireless signal in the 5.7 GHz band. Two notebook computers with Lenovo e531 CPU Intel core i7 3632QM, operating system Windows 10, and equipped with Xshell 5.0 and WinSCP2.0 software are used as operation entry devices to connect transmit and receive routes.

In this paper, the performance of the Wi-SL method is verified in three classical indoor scenarios. According to the multi-path effect from strong to weak, the experimental environment is set to a laboratory (8 m × 10 m) containing more desks, computers, and other furniture, a more empty corridor (3 m × 8 m) and an empty hall (12 m × 15 m). The experimental scenarios and the scenarios plan structure are shown in Figure 13 and Figure 14, respectively.

We deployed routing equipment in the above three experimental scenarios and arranged 5 experimenters (3 males and 2 females) aged 23–26 who were accustomed to using the right hand to perform sign language actions in the middle of the two routers, collected their gesture CSI data, and used 70% of this data as a dataset for offline fingerprint training, and the remaining 30% as a test set for verification. We asked the experimenters to only move their hands and arms and keep other body parts intact during sign language gestures. The distance between Tx and Rx is set to 1.5 m, and the height of Tx and Rx is set to 1.2 m. This setting takes into account the height of human gestures and the relatively short distance in the actual scene to obtain a large signal-to-noise ratio of wireless signals. Set the packet sending rate to 50 packet/s, sample 5000 packets at a time, each sign language action collection lasts about 2 min, the Finger language gestures are kept in the 5 × 5 cm range, and the Sign language gestures are kept in the 30 × 30 cm range. This setting is reasonable, taking into account the size of each hand and the habitual range of actions. In the course of the experiment, the user’s information is tested, and the user’s sign language sampling times and sampling duration are shown in Table 1.

In Table 2, we give the relevant flow of Wi-SL for sign language action recognition, and the specific hardware used in each step, and quantify the specific time required for the hardware to process these operations. Although it takes only 1.5 s to identify an action when an offline fingerprint has been established, the time spent on the establishment of the fingerprint in the early stage and the timeliness in the actual scene are still insufficient. We will shorten the recognition time in future work and improve the practicability.

### 4.2. Experimental Analysis

#### 4.2.1. Impact of LOS/NLOS Scenarios

In the LOS and NLOS scenarios, wireless signals will form different reflections and refraction paths for sign language actions, which will affect the result of gesture recognition. Therefore, we compare the sign language actions in the laboratory, corridors, and halls with the influence of LOS and NLOS scenarios. The comparison results are shown in Figure 15, where Figure 15a shows 6 Finger language gestures affected by LOS and NLOS scenarios, and Figure 15b shows 6 Sign language gestures affected by LOS and NLOS scenarios.

First, it can be seen from Figure 15a,b that the sign language recognition effect in the LOS scenario is better than that in the NLOS scenario. Because LOS is a direct path, the influence of the human hand on the wireless channel can be better expressed in the CIR data received by Rx. NLOS is a non-direct path, and there are reflections of the hand and multipath factors, which will reduce the recognition accuracy. However, in actual sign language application scenarios, LOS is a more ideal state, and NLOS is more close to practice. Finally, through calculation and analysis, the average recognition rate of Wi-SL in three experimental scenarios under LOS can reach 95.8%, and the average recognition rate under NLOS can reach 89.3%.

#### 4.2.2. The Impact of User Diversity and Sign Language Range

To explore the impact of user diversity on Wi-SL sign language recognition, 5 testers were set up, of which User1, User2, and User3 were tested in the LOS scenario, and User4 and User5 were tested in the NLOS scenario, keeping other experimental conditions consistent. Test all the sign language actions selected in this paper in 3 experimental scenarios. In addition, we also want to understand the effect of the sign language writing range on Wi-SL detection. According to the range of Sign language gesture and Finger language gesture selected in this article, they are tested separately. There are two action ranges of 5 × 5 cm and 10 × 10 cm for the Finger language gesture, and two action ranges of 30 × 30 cm and 40 × 40 cm for the Sign language gesture. Use 120 samples of data generated by 5 testers in the same experimental environment for training and analysis. The experimental results of these two tests are shown in Figure 16.

As can be seen from the (a) in Figure 16, Wi-SL has always reliably detected the sign language actions of five experimental users, who differed not only in terms of sex and age, but also in arms length and hand size due to height and weight, especially for User4 and User5, even under the condition of NLOS, the average detection rate of Wi-SL is more than 87%. Figure 16b shows the influence of two different action ranges of the Finger language action recognition, while Figure 16c shows the influence of two ranges of the Sign language action recognition, from which we can see that the average accuracy of the Finger language action has increased from 91.27% to 93.03%, and the average accuracy of the Sign language action has increased from 93.64% to 95.25%. Therefore, it can be concluded that the effect of sign language recognition increases with the increase of sign language action range. This is because CSI has good mobility sensitivity. When the amplitude of the user’s action becomes larger, CSI can better describe the impact of sign language action on the channel. However, in the actual sign language use scene, the action range of sign language is usually small. Wi-SL has achieved good detection results for both small action range and large action range, which meets the needs of actual sign language recognition.

#### 4.2.3. The Impact of Channel Frequency and Bandwidth

The channel bandwidth setting affects the communication frequency band of the wireless signal. The frequency of the wireless signal determines its penetration ability and propagation path loss [42]. The channel settings in the experiments in this paper are shown in Table 3.

Therefore, to analyze the influence of channel setting on the experiment, we test the sign language action when the bandwidth is 20 MHz frequency 2.4 GHz and the bandwidth 40 MHz frequency is 5.7 GHz, and ensure that other experimental conditions are consistent. The impact of using different channel parameters to set the results of sign language recognition is shown in Figure 17.

The (a) and (b) in Figure 17 are time-domain amplitude diagrams of VERY sign language actions under the conditions of 2.4 GHz and 5.7 GHz. Different colors in the time-domain amplitude diagram represent different values of CSI amplitude, blue represents a smaller value, and yellow represents a larger value. The influence of sign language action on wireless signal is shown as the change of energy in the frequency domain, specifically the magnitude of the amplitude.

We can see that less yellow high-frequency data are showing the influence of human hand movements under 2.4 GHz conditions, while more yellow high-frequency data are reflecting human hand movements under 5.7 GHz conditions. There are two reasons for this. First, since the subcarrier index of 5.7 GHz under the IEEE 802.11n protocol is 1 to 114, it can provide more frequency domain information of the subcarrier level than the subcarrier index 1 to 56 of 2.4 GHz. Second, compared with the 2.4 GHz wireless channel, the wireless channel of 5.7 GHz will not be occupied by other commercial wireless devices of the environment. The increase in the center frequency can greatly enhance the sensitivity to channel change perception. Specifically, it can sensitively perceive small changes in hand movements. Combined with the comparison result in Figure 17c, the sign language recognition effect with a bandwidth of 20 MHz and a frequency of 2.4 GHz is lower than that of 40 MHz and a frequency of 5.7 GHz. It can be concluded that Wi-SL works well under the channel setting with a bandwidth of 40 MHz and a frequency of 5.7 GHz.

#### 4.2.4. The Influence of Distance and Personnel Interference in NLOS Scenario

Most of the application scenarios of gesture recognition belong to NLOS scenarios, to better explore the effect of gesture recognition in NLOS situations. We set up distance factors and interference factors to complete the experimental analysis. First of all, we test the distance factor, that is, keep the other experimental settings unchanged in the same experimental scene, and let the testers perform gestures at different distances (1 m, 2 m, 3 m, 4 m, 5 m) according to the NLOS in Figure 14. The specific gesture recognition effect is shown in Figure 18a. And then, for the interference factors, we let the testers perform gestures in the NLOS situation by keeping the other experimental conditions fixed, while arranging another tester to walk normally at a distance of 3.5 m as interference. We compared the effect of gesture recognition with and without human interference, as shown in Figure 18b.

We can see clearly from Figure 18a that when the distance is 2 m, better gesture recognition results are obtained in the case of NLOS. When the distance is more than 2 m, with the increase of signal reflection and scattering, the effect of gesture recognition continues to decline, but as a whole, when the distance is 1–4 m, the gesture recognition rate of Wi-SL is more than 80%, maintaining a good performance. However, in Figure 18b, even in the presence of human interference, Wi-SL achieves an average gesture recognition rate of 82.4%, indicating that our system can complete gesture recognition even if there is human interference. However, in terms of practical application scenarios, there is a large room for improvement in the Wi-SL system, and we will continue to improve and improve the performance of the system in the next work.

### 4.3. Overall Performance Evaluation

#### 4.3.1. Robustness Analysis

To verify and analyze the robustness of Wi-SL, we compare Wi-SL with the existing three CSI gesture recognition work (WiGeR, WiAG, WiG) in terms of environmental adaptability and usability. Firstly, for the environmental adaptability of different systems, in the three experimental scenarios constructed in this paper, keeping the same experimental conditions, 12 sign language action data are collected, and the detection accuracy of Wi-SL is compared with that of the other three methods. The results of comparison and analysis are shown in Figure 19a. Secondly, we analyze the availability of the system by comparing the detection range of different systems, and in the experimental scenario constructed in this paper, the control variable method is used to keep other experimental conditions consistent, so that the testers performed 12 sign language actions at a distance of 0.5 m, 1 m, 1.5 m, 2 m, 2.5 m, 3 m, 3.5 m from the transceiver device, and processed these sign language data using Wi-SL and WiGeR, WiAG, WiG, respectively. Finally, the specific results are shown in Figure 19b.

As shown in Figure 19a, with the change of environment, the accuracy of sign language gesture recognition of each method is improved due to the weakening of the multipath effect. Compared with the WiGeR, WiAG and WiG methods, Wi-SL has achieved average recognition accuracy of not less than 91% in the laboratory, corridor, and hall, and has achieved an impressive 95.8% in the hall. Therefore, Wi-SL has good adaptability (robustness) to the environment. As shown in Figure 19b, with the increase of the distance between the user and the transceiver equipment, the recognition accuracy of Wi-SL decreases slightly, but it can still reach the average accuracy of 94.7%, while WiGeR, WiAG, and WiG can achieve the average recognition accuracy of 93%, 91.2%, and 92.4%, respectively. Considering the actual application scenario, the deployment of transceiver devices should be far away from users. In comparison, the availability of Wi-SL is stronger and more stable.

#### 4.3.2. Classification Effect Evaluation

In this section, to demonstrate the classification performance of the KSB model in Wi-SL, we define four metrics as shown in Table 4. These metrics are used to evaluate the classification performance of methods such as Wi-SL and WiGeR, WiAG, WiG. First of all, the ROC curve (used to evaluate the generalization ability and classification efficiency of the classification method) is introduced, where the threshold is the default 0.5, and 10000 CSI packet data are used to analyze the classification effect of the KSB model. Secondly, we use Precision, Recall, and F1−Score in Table 4 to evaluate the comprehensive performance of Wi-SL and the existing three methods. Finally, the results are shown in Figure 20.

Here, TPR=Ture Positives Rate, TNR=Ture Negatives Rate, TP=Ture Positives, TN=Ture Negatives, FP=False Positives, FN=False Negatives.

The ROC curve analysis of Wi-SL is shown in Figure 20a. It can be seen that when the false detection rate is 10%, the correct gesture detection rate can reach 92.08%. This alone is not enough to show that the classification effect of Wi-SL is excellent. Therefore, we compare the classification performance of Wi-SL with the three methods of WiAG, WiGeR, and WiG. Figure 20b shows specific metric evaluation results. It can be seen that the Precision, Recall, and F1−Score indicators of Wi-SL are significantly better than the other three methods. There are two reasons for such results: The stability of selected features. (2) The computational complexity of the classification algorithm.

First of all, Wi-SL uses the phase difference and amplitude of the subcarrier level extracted by PCA as recognition features, which can stably reflect the influence of fine-grained gestures on wireless signals, and greatly reduces the computational complexity of gesture features The KSB classification model makes the feature data go through K-Means clustering and then classifies by the optimal SVC learned by the Bagging algorithm to obtain high-precision and reliable classification results. However, WiGeR uses a single amplitude data as a gesture recognition feature, and only uses the DTW algorithm. Although it has good recognition accuracy, the DTW algorithm requires a lot of calculation costs. Secondly, WiAG uses a time-domain amplitude as a gesture feature. In addition, it uses the k-Nearest Neighbors (KNN) algorithm for data classification. It does not form a classification model but only calculates the distance between feature data to achieve classification, which causes a high calculation burden and low classification accuracy. Finally, the WiAG system uses the Local Outlier Factor (LOF) as a gesture recognition feature, which is easily changed by the environment. A single SVM method is used in the classification method, and the overall effect is general. Through comparison and analysis, we have effectively proved the stability of the Wi-SL system classification model and the higher classification accuracy and efficiency.

#### 4.3.3. Comparison with Existing Technologies

To reflect the comprehensive performance of the Wi-SL method, we analyze the Wi-SL method and the existing three types of gesture recognition methods (Computer Vision, Wearable Sensors, Radio equipment) in five dimensions: intrusiveness, deployment cost, device-free, system complexity, and detection accuracy. The comparison results are shown in Table 5.

As can be seen from Table 5, the Wi-SL method simultaneously meets the requirements of non-invasion, low deployment cost, device-free, and low system complexity. The other three methods have high deployment costs and system complexity. Among them, the Computer Vision method will cause a certain degree of intrusiveness and cannot work in the private environment. However, wearable sensor and radio equipment methods require professional and complex equipment, and the cost of use and maintenance is high. Although the accuracy of gesture recognition is very high, it is not suitable for large-scale popularization.

## 5. Conclusions

In this paper, we propose a fine-grained sign language gesture recognition method, Wi-SL, which uses channel state information, non-invasive and high robustness to achieve high-precision recognition of 12 gestures. Wi-SL constructs a powerful denoising method and a reasonable classification method, using CSI signals to efficiently perceive the fine-grained hand movements of the target, and analyzes the impact of LOS/NLOS scenarios, user diversity and gesture writing range, channel frequency and bandwidth on the accuracy of gesture recognition, the performance of Wi-SL was evaluated in three different scenarios, and the classification performance of KSB classification model in Wi-SL was analyzed by some comprehensive measures. The experimental results show that the average accuracy of Wi-SL gesture detection has reached 95.8%. Therefore, Wi-SL has great potential to become a practical and non-invasive gesture recognition solution. In addition, we will further improve the recognition accuracy in future work and consider the recognition research on the sign language of the two-handed type.

## Figures and Tables

**Figure 1 sensors-20-04025-f001:**
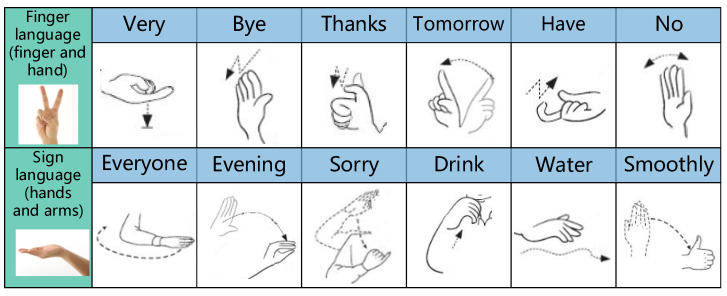
Twelve commonly-used sign language diagrams.

**Figure 2 sensors-20-04025-f002:**
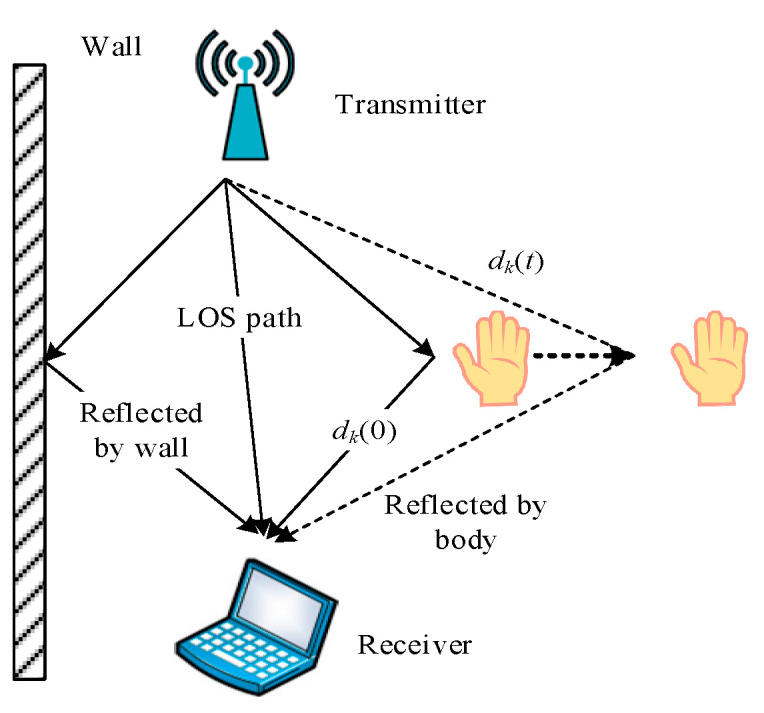
The influence of human hand movements on wireless signals.

**Figure 3 sensors-20-04025-f003:**
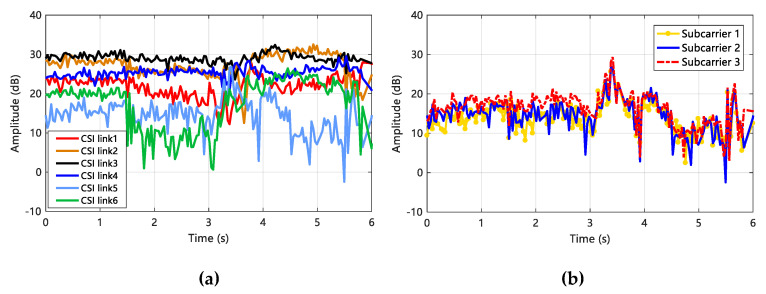
The influence between Channel State Information (CSI) data link and subcarrier amplitude. (**a**) Subcarrier amplitude distribution in different CSI data links; (**b**) amplitude distribution of different subcarriers in the same CSI data link.

**Figure 4 sensors-20-04025-f004:**
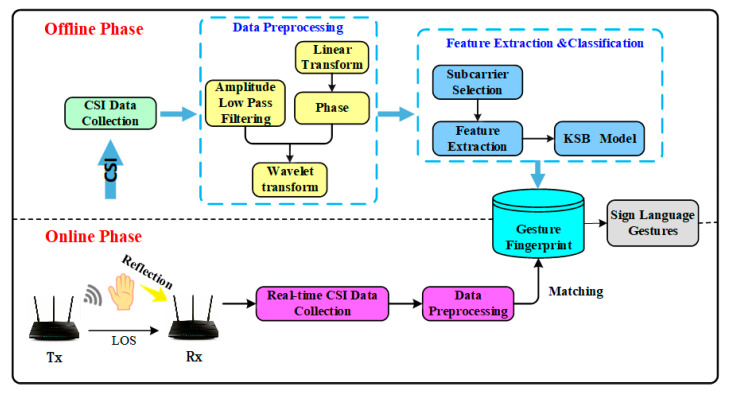
Wi-SL method flow chart.

**Figure 5 sensors-20-04025-f005:**
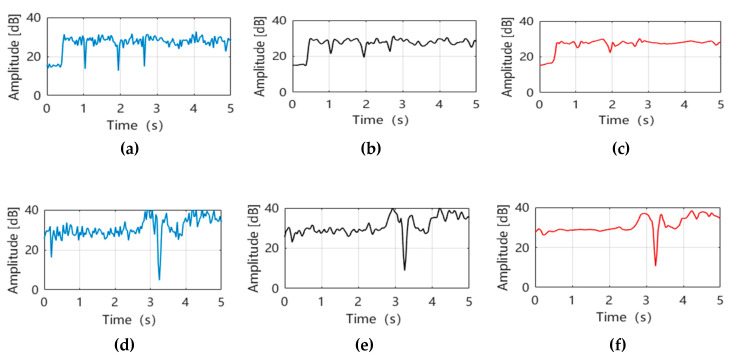
Amplitude data outlier filtering. (**a**) Original Sign language data 1; (**b**) Sign language data 1 is processed by Butterworth low-pass filtering; (**c**) Sign language data 1 smoothed by wavelet function; (**d**) original sign language data 2; (**e**) Sign language data 2 is processed by Butterworth low-pass filtering; (**f**) Sign language data 2 smoothed by wavelet function.

**Figure 6 sensors-20-04025-f006:**
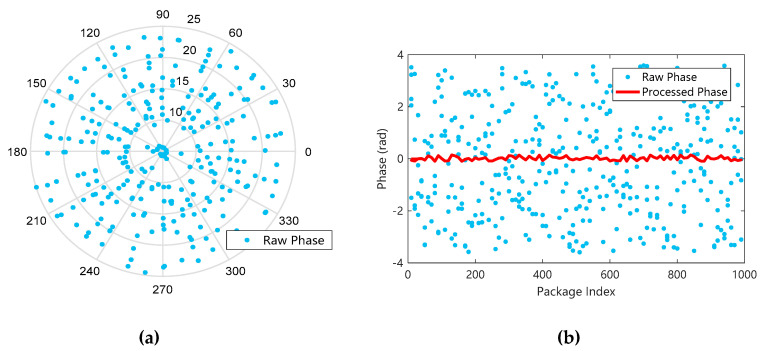
Raw phase and phase processed by linear transformation. (**a**) Randomly distributed raw phase data; (**b**) phase data after linear transformation.

**Figure 7 sensors-20-04025-f007:**
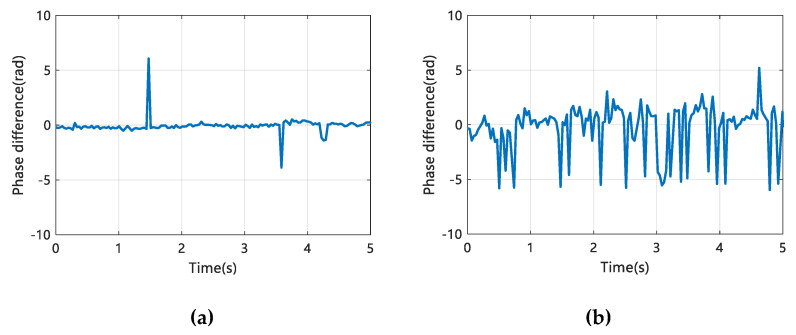
The influence of sign language action on phase difference. (**a**) Sign language gesture; (**b**) Finger language gesture.

**Figure 8 sensors-20-04025-f008:**
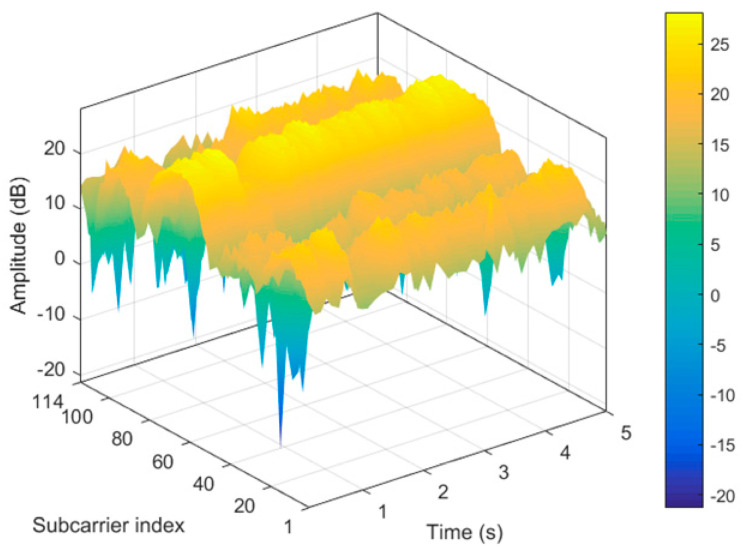
The amplitude of 114 subcarriers of sign language gestures are transformed in the time domain.

**Figure 9 sensors-20-04025-f009:**
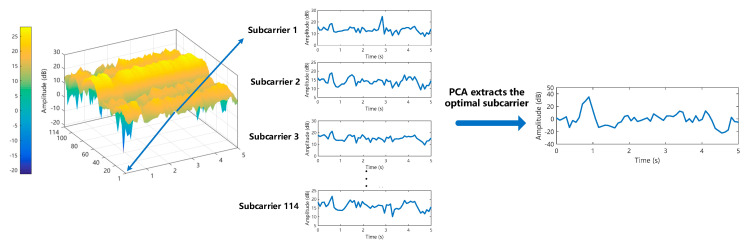
Principal components analysis (PCA) extracts the optimal sub-carrier among 114 sub-carriers.

**Figure 10 sensors-20-04025-f010:**
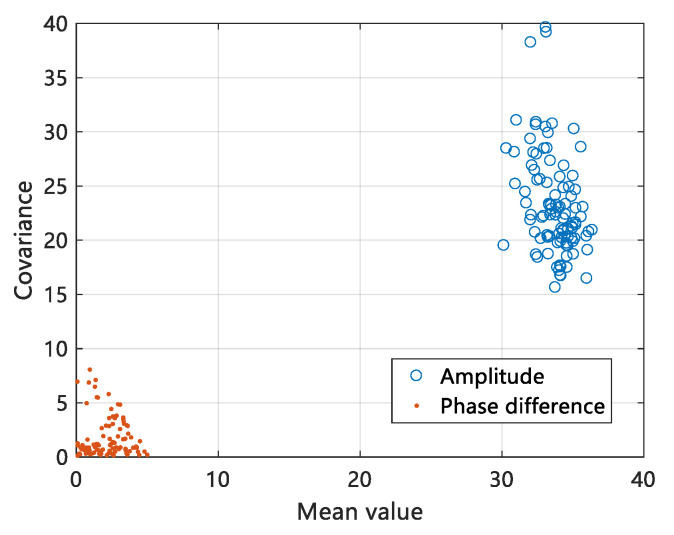
Independent verification of amplitude and phase difference data.

**Figure 11 sensors-20-04025-f011:**
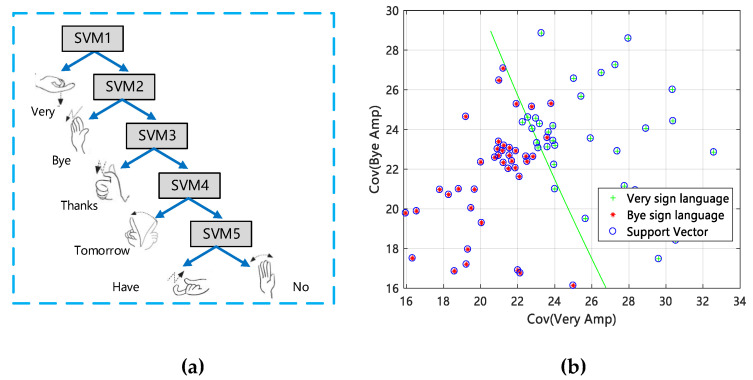
Support Vector Machine (SVM) classification. (**a**) SVM gesture classification decision tree model; (**b**) preliminary classification result using an SVM.

**Figure 12 sensors-20-04025-f012:**
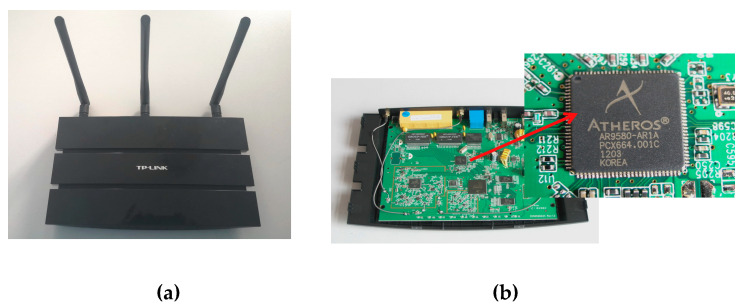
Experimental hardware equipment. (**a**) TP-Link WDR4310 router; (**b**) Atheros AR9580 network card chip.

**Figure 13 sensors-20-04025-f013:**
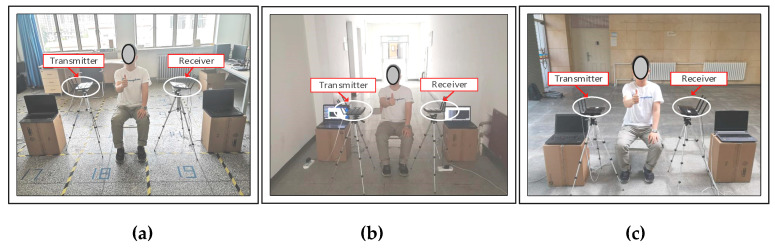
Three experimental scenarios: (**a**) laboratory; (**b**) corridor; (**c**) hall.

**Figure 14 sensors-20-04025-f014:**
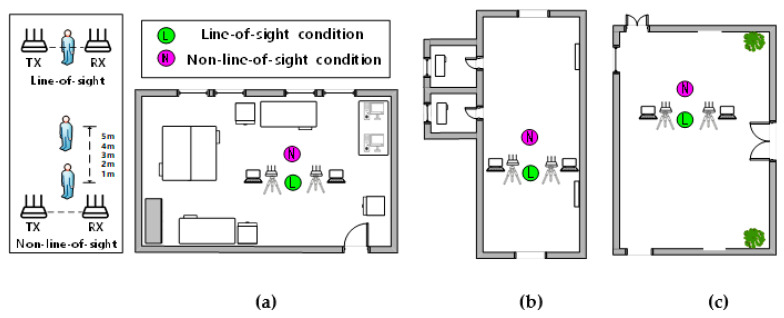
Floor plan of the experimental scenarios: (**a**) laboratory; (**b**) corridor; (**c**) hall.

**Figure 15 sensors-20-04025-f015:**
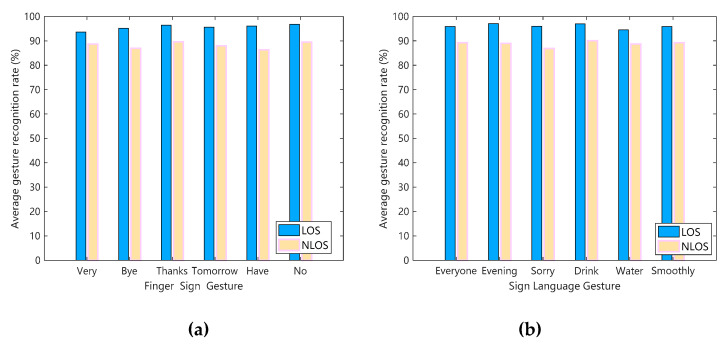
Effects of Line-of-Sight (LOS) and Non-Line-of-Sight (NLOS) scenarios on sign language actions. (**a**) The Finger language gesture; (**b**) the Sign language gesture.

**Figure 16 sensors-20-04025-f016:**
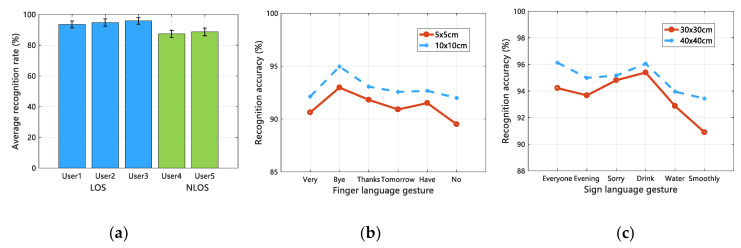
The influence of user diversity and sign language action range. (**a**) User diversity analysis; (**b**) Finger language range comparison; (**c**) Sign language range comparison.

**Figure 17 sensors-20-04025-f017:**
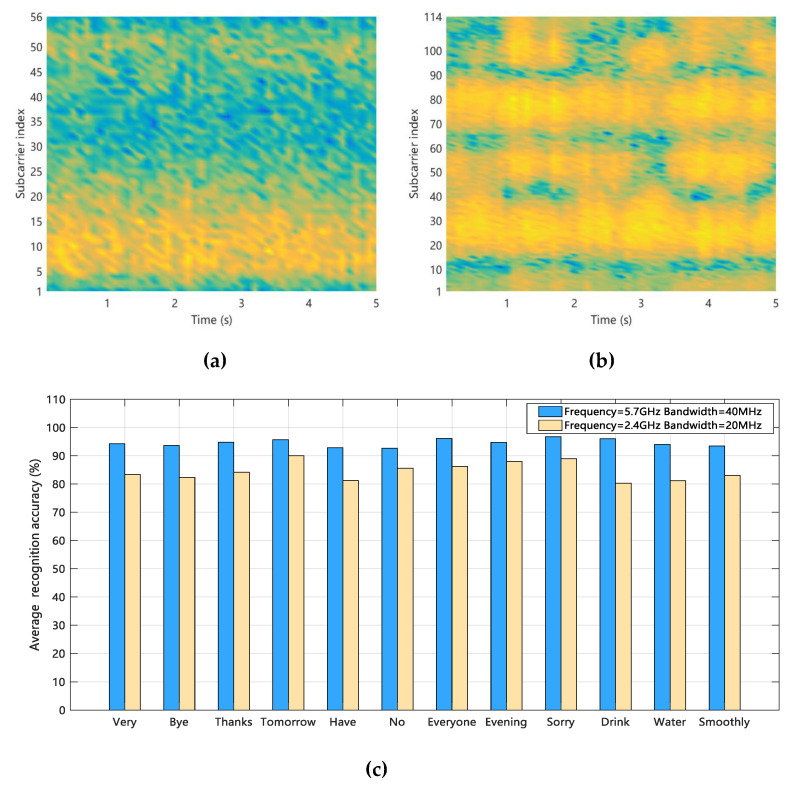
Effects of different channel parameter settings on sign recognition results. (**a**) 2.4 GHz time-domain amplitude diagram; (**b**) 5.7 GHz time-domain amplitude diagram; (**c**) comparison of sign language recognition effects under two kinds of channel parameter settings.

**Figure 18 sensors-20-04025-f018:**
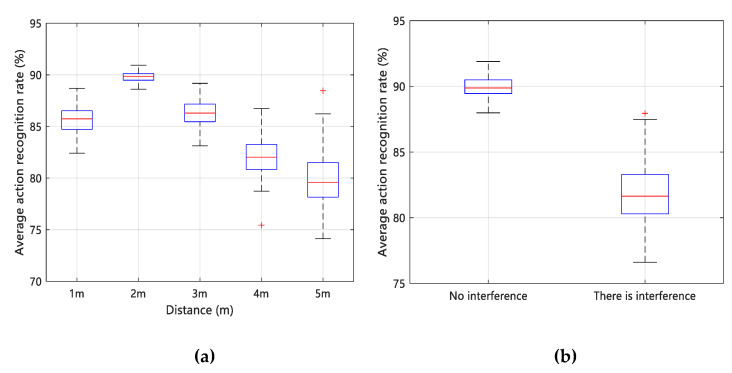
Influence of distance and personnel interference on gesture recognition effect under the NLOS scenario. (**a**) Distance; (**b**) personnel interference.

**Figure 19 sensors-20-04025-f019:**
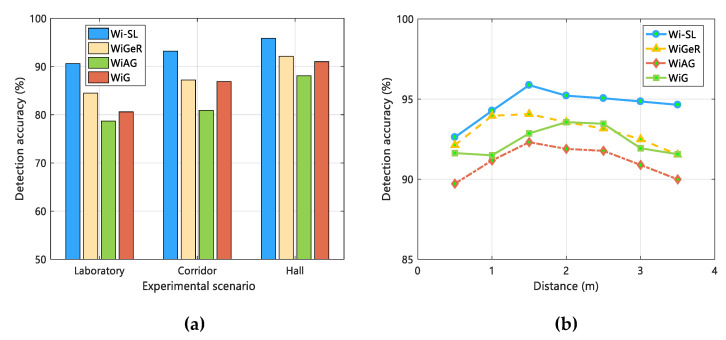
Robustness comparison between Wi-SL and the other three methods. (**a**) Environmental adaptability analysis; (**b**) detection distance analysis.

**Figure 20 sensors-20-04025-f020:**
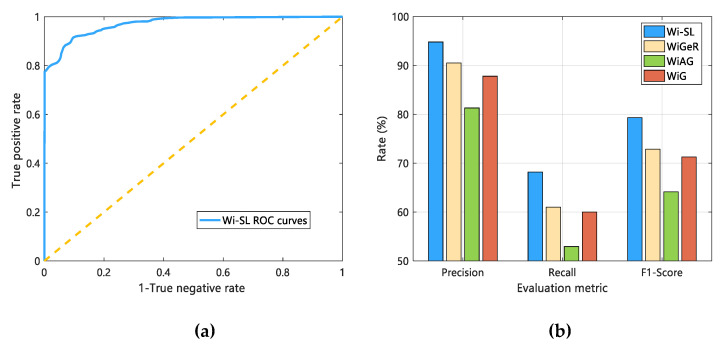
A comprehensive evaluation of classification performance. (**a**) ROC curve of Wi-SL; (**b**) Comprehensive evaluation of 4 methods.

**Table 1 sensors-20-04025-t001:** Sign language data collection summary (G1 = VERY, G2 = BYE, G3 = THANKS, G4 = TOMORROW, G5 = HAVE, G6 = NO, G7 = EVERYONE, G8 = EVENING, G9 = SORRY, G10 = DRINK, G11 = WATER, G12 = SMOOTHIY).

			Data Collection		
User Age/Sex		Ht/Wt	Gestures Collected	Collection Times	Duration
User1	26/Male	75 kg/176 cm	G1–G12	5	1.67 h
User2	25/Male	80 kg/180 cm	G1–G6	3	0.50 h
User3	23/Male	95 kg/187 cm	G1–G12	8	2.67 h
User4	23/Female	48 kg/162 cm	G1–G12	4	1.33 h
User5	24/Female	55 kg/170 cm	G7–G12	2	0.33 h

**Table 2 sensors-20-04025-t002:** Sign language data collection summary.

	Data Collection	Offline Phase	Online Recognition
Hardware type	TP-Link router	Lenovo laptop	TP-Link router + Lenovo laptop
Processing time	102.52 s	12.01 s	1.21 s

**Table 3 sensors-20-04025-t003:** Experimental channel settings.

Bandwidth	Channel Frequency Segment	Number of Sub-Carriers	Index of Sub-Carriers
20 MHz	2.412~2.472 GHz	56	(−28, −27,...,−2,−1,1,2,...,27,28)
40 MHz	5.725~5.825 GHz	114	(−58, −57, −56,...,−3,−2,2,3,...,56,57,58)

**Table 4 sensors-20-04025-t004:** Different measures of overall performance analysis.

Definition of Metrics	Also Defined	The Role of Metric
Precision=TP/(TP+FP)	Precision	The probability of accurate detection of gesture detection.
TPR=Recall=TP/(TP+FN)	Sensitivity	Correctly identify the probability of detecting gestures in the scenario.
TNR=TN/(TN+FP)	Specificity	Correctly identify the probability of undetected gestures in the scenario.
F1−Score=2*Precision*RecallPrecision+Recall	F1−Score	A comprehensive index that can effectively evaluate the stability of the method.

**Table 5 sensors-20-04025-t005:** Comparison of capabilities between Wi-SL and various gesture recognition methods.

Parameters	Mtheods			
	Wi-SL	Computer Vision	Wearable Sensors	Radio Equipment
Intrusive or privacy	No	Yes	Yes	No
Deployment costs	Low	Medium	High	High
Device-free	Yes	No	No	No
System complexity	Low	High	High	Medium
Detection accuracy	High	High	High	High

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
