# Peer review of "Wi-SL: Contactless Fine-Grained Gesture Recognition Uses Channel State Information"

_sensors, 2020, doi:10.3390/s20144025_

Round 1

Reviewer 1 Report

This paper has introduced a WiFi CSI-based gesture recognition sysytem which reaches an accuracy of 95.8%. Although the performance is favorable, there are severla problems to be handled before it is accepted for publication.

First of all, the motivation is not strong enough. WiFi CSI-based gesture recognition techniques have been studied for several years and many systems have been proposed on top conferences such as Ubicomp, Infocom, Sensys,Mobicom and etc. These works focueses on dealing with different problems related with CSI-based gesture recognition and covers a wide area of applications. As a result, I am not well convinced by the motivation of this paper. That is, what are the unqiue contributions of this work? The authors said that they make use of the different phase difference information produced by the motion of different parts of the hand. But the pahse information of CSI have been also utilized by other groups such as Prof. Daqing Zhang in Peking University for gesture/activities recognition. So I suggest that the authors polish the introduction part to demonstrate the motivation more clearly.

Second, the techniques used in the paper seem to lack novelty. The claimed controbutions of techniques in the introduction part such as denosing methods, classification models seem to be not so significant. And there lacks explanation of the rationale and necessarity of adopting these methods. 

Third, as for the evaluation of user diversity, I am curious about the where the training data and testing data come from respectively. In other words, the authors should describe more clearly about how the classification model is trained and tested. This is because when the training and testing data are from different experimenters, the performance may be rather different from the that of the case where training data and testing data are from the same person.

Fourth, the evaluation of impact of NLOS seems to be too simple. I suggest the authors add experiments where the experimenter stays at different positions and distances in the NLOS case, to evaluate the impact of NLOS more comprehensively. In this way, the evaluation of system robustness can be more convinced.

Author Response

Thank you for your suggestions, we have made a serious reply to each of your suggestions, please refer to the attached file for the specific reply.

Reviewer 2 Report

The authors present a contactless fine-grained gesture recognition method using Wi-Fi Channel State Information. This paper has enough content to consider publishing, but there are some problems that the author should solve in the revision.

More specifically, the paper has the following strong points:

  1. The presentation of the paper is of high quality, as figures are included presenting all the necessary aspects of the system and the scenarios.
  2. The preliminaries provide the background knowledge of Channel State Information and gesture feature selection.
  3. The detailed flow is given in the design phase of the system, and all its components are fully described.
  4. The configuration of the experiment is given in detail, which is convenient for readers to understand the necessary details of the experiment and the parameters of the study.
  5. The sufficient detailed description of the experiment and the results obtained are provided, and meaningful conclusions are drawn by analyzing the factors affecting the experiment and the overall performance of the system.

Still, the following aspects should be revised:

  1. In the process of data processing, please give the relevant technical parameters and theoretical analysis in detail.
  2. Please check whether all abbreviations meet the academic specifications, for example k-Nearest Neighbor should be k-NN.
  3. Please check whether the in Formula 4 and Formula 11 represent the same parameter, if not, keep one and replace the other.
  4. The theoretical analysis is relatively weak. Please make a further discussion or set up a concrete model to address why the collected signal could effectively recognize the gesture. Although the data set is sufficient, the reviewer and potential readers still want to know the basic rationale behind it.
  5. Please add some interference in the system to see the robustness of this work. For example adding the interfering devices and interfering human into the system to check the stability of the system.
  6. Please further discuss some limits of the system. To make the working condition of the system more clear, the reviewer needs to see the performance boundary of the system.

Author Response

(The authors gave the same response as above.)

Reviewer 3 Report

see attached file

Author Response

(The authors gave the same response as above.)

Round 2

Reviewer 3 Report

See attached file.

Author Response

(The authors gave the same response as above.)
